# Healthcare Worker Characteristics Associated with SARS-CoV-2 Vaccine Uptake in Ireland; a Multicentre Cross-Sectional Study

**DOI:** 10.3390/vaccines11101529

**Published:** 2023-09-26

**Authors:** Liam Townsend, Gavin Kelly, Claire Kenny, Jonathan McGrath, Seán Donohue, Niamh Allen, Lorraine Doherty, Noirin Noonan, Greg Martin, Catherine Fleming, Colm Bergin

**Affiliations:** 1Department of Infectious Diseases, St James’s Hospital, D08 NHY1 Dublin, Ireland; townsenl@tcd.ie (L.T.);; 2Department of Infectious Diseases, University Hospital Galway, H91 YR71 Galway, Ireland; 3Health Protection Surveillance Centre, D01 A4A3 Dublin, Ireland; 4Department of Occupational Medicine, St James’s Hospital, D08 NHY1 Dublin, Ireland; 5Department of Clinical Medicine, Trinity College Dublin, D02 PN40 Dublin, Ireland

**Keywords:** vaccination, healthcare workers, COVID-19, prevention

## Abstract

The prevention of SARS-CoV-2 acquisition and transmission among healthcare workers is an ongoing challenge. Vaccination has been introduced to mitigate these risks. Vaccine uptake varies among healthcare workers in the absence of vaccine mandates. We investigated engagement with SARS-CoV-2 vaccination among healthcare workers and identified characteristics associated with lower vaccine uptake. This multi-site cross-sectional study recruited n = 1260 healthcare workers in both clinical and non-clinical roles over a three-month period from November 2022. Participants reported their engagement with the primary SARS-CoV-2 vaccination programme and subsequent booster programmes, as well as providing demographic, occupational and personal medical history information. Multivariable linear regression identified characteristics associated with vaccine uptake. Engagement with vaccination programmes was high, with 88% of participants receiving at least one booster dose after primary vaccination course. Younger age and female sex were associated with reduced vaccine uptake. Healthcare workers in non-clinical roles also had reduced vaccine uptake. These findings should inform vaccination strategies across healthcare settings and target populations with reduced vaccine uptake directly, in particular young, female, and non-clinical healthcare workers, both for SARS-CoV-2 and other healthcare-associated vaccine-preventable infections.

## 1. Introduction

Severe Acute Respiratory Syndrome Coronavirus-2 (SARS-CoV-2) is responsible for the global COVID-19 pandemic. The characteristics of this pandemic have evolved since the virus was first identified in late 2019. The initial B.1.1.7 alpha variant is no longer the circulating variant of concern, with subsequent variants varying in the severity of illness caused and rates of onward transmission [1]. The emergence of new variants of concern has driven sequential waves of infection, with peaks of symptomatic infection [2,3]. Healthcare workers (HCWs) are at increased risk of SARS-CoV-2 acquisition compared to the general population [4,5]. Prevention of HCW infection is important to reduce workforce illness, reduce onward transmission to patients, and limit employee absenteeism, as well as mitigating the risks of developing persistent ill health in the form of long COVID [6,7,8].

Multiple methods have been used to reduce transmission of SARS-CoV-2 among HCWs, including use of personal protective equipment and limitations on time spent in direct contact with COVID-19 patients [9,10]. The development of vaccines against SARS-CoV-2 has been one of the tools employed to mitigate HCW infection [11,12,13], with proven efficacy in HCW populations [14]. Vaccine mandates were introduced in some jurisdictions to ensure adequate HCW vaccination; however, this was not the case in the Republic of Ireland [15]. The Irish national vaccination strategy was initiated in December 2020, with primary vaccination taking place over the following months. A booster vaccination programme for HCWs was rolled out in November 2021. A month prior to this, boosters were available to immunocompromised individuals and those older than 80. Subsequent booster vaccinations for HCWs were recommended from July 2022 [16]. In the absence of vaccine mandates, uptake of vaccination among HCWs has varied, with factors including male sex, clinical role, and age previously having been found to be associated with vaccine acceptance [17,18]. The importance of primary vaccination and ongoing engagement with booster vaccination programmes is evidenced by the relatively poor protection against infection offered by the primary vaccination course against new variants of concern, such as Omicron [19]. There has been an evolution in the public’s perception of receiving COVID-19 vaccinations over the course of the pandemic in Ireland, with increasing resistance and hesitancy being seen [20,21]. This ongoing engagement continues to pose a challenge to vaccination teams and occupational health departments. 

The Prevalence of Antibodies to SARS-CoV-2 in Irish HCWs (PRECISE) study is a multicentre cross-sectional study of COVID-19 vaccine uptake and antibody response in HCWs [22,23,24]. Here, we investigate the demographic and epidemiological factors associated with primary vaccine uptake and receipt of additional booster vaccines in this population. We aim to identify HCW characteristics associated with reduced vaccine uptake in order to inform future vaccination campaigns for both SARS-CoV-2 and other vaccine-preventable conditions such as Influenza.

## 2. Materials and Methods

### 2.1. Study Setting and Participants

This was a multi-site cross-sectional study carried out in two hospital sites in Ireland, namely, St James’s Hospital (SJH) in Dublin and University Hospital Galway (UHG). Data collection took place over a period from November 2022 to January 2023. Both study sites are tertiary referral centres, with SJH having approximately 4700 staff and UHG having approximately 4400 staff. All employees at both hospital sites were invited to participate via internal hospital communications, emails and text messages. Ethical approval for the study was obtained from the local ethics committees, application numbers GCREC 15/09/2022 C.A. 2860 and TUH/SJH REC 2022-Nov-23002300. Written informed consent was obtained from all participants.

Inclusion criteria were any HCW whose primary site of employment was either SJH or UHG. HCWs were defined as any person employed to work on-site in the hospital, and was not limited to those with direct patient contact. All participants were 18 years of age or older. Participants were excluded if they did not have an email address or were not suitably proficient in English to complete the survey.

### 2.2. Covariate Assessment

Participants completed an online questionnaire, with information regarding demographics, staff role, SARS-CoV-2 vaccination history, and personal risk factors for SARS-CoV-2 infection including past medical history recorded. Staff role was divided into clinical and non-clinical roles. Clinical roles incorporated medical/dental, nursing/midwifery, healthcare assistants, allied health professionals and laboratory workers, while non-clinical roles included administrative staff, portering and transport, security, catering, and maintenance. Allied health professionals incorporated physiotherapists, occupational therapists, social workers, dieticians, speech and language therapists, and radiographers. The personal risks for COVID-19 infection assessed were receipt of biologic therapy, rituximab, methotrexate or steroids (40 mg daily for one week or 20 mg daily for two weeks), as well as prior organ transplantation or active malignancy. Access to personal medical records was not undertaken; personal risks were self-declared. All vaccinated study participants received their COVID-19 vaccine as part of a two-dose regimen with either the Comirnaty (Pfizer/BioNTech, Mainz, Germany) vaccine, the Vaxzevria (AstraZeneca, Sodertalje, Sweden) vaccine, the Moderna vaccine (Madrid, Spain, or the Janssen/Johnson & Johnson vaccine (Beerse, Belgium). Four SARS-CoV-2 vaccination statuses were recorded: 1. Incomplete primary vaccination course/unvaccinated; 2. Primary vaccine course only; 3. Primary vaccine course and one booster; 4. Primary vaccine course and at least two boosters. Vaccination history was confirmed by cross-checking self-reported vaccination status with COVAX, the national COVID-19 immunisation system.

Location of employment was used as a covariate, i.e., located in SJH and located in UHG. This was used to account for differences in approaches to vaccination taken at both sites. All vaccine doses were administered on the main hospital campus in SJH, while primary vaccinations were administered on the main hospital campus and subsequent boosters at a satellite campus in UHG.

### 2.3. Statistical Analysis 

All analysis was performed using Stata version 18.0 (Stata Statistical Software, College Station, TX StataCorp LP). Statistical significance was indicated by *p* < 0.05. Descriptive statistics are reported as means with standard deviations (SD) and medians with interquartile ranges (IQR), as appropriate. Univariate analysis was performed on demographic variables to examine the associations with receiving a full vaccination course using *t*-test (t), Wilcoxon rank-sum test (z), Chi-squared test (χ^2^) and analysis of variance tests. Tukey or Dunne testing was performed post hoc. Multivariable stepwise logistic regression including all univariate variables, with fully vaccinated as the dependent variable, was used to analyse factors associated with completing the vaccination course. Adjustments to the regression model were made for age, sex, location of employment (with located in SJH as the reference), country of birth (Ireland vs. non-Ireland), ethnicity, self-reported acquisition risk, and clinical vs. non-clinical role. The stepwise regression model was instructed to eliminate variables with *p* values > 0.20, in keeping with best practice [25]. Models were examined for multicollinearity by computing variance inflation factors and visually examining residual-versus-fit models. Results are reported as odds ratios with 95% confidence intervals and corresponding *p* values.

## 3. Results

### 3.1. Cohort Descriptors

A total of 1260 HCWs took part, representing 14% (1260/9100) of the total employee cohort of both sites, with n = 640 (51%) at SJH and n = 620 (49%) at UHG. The characteristics of the total cohort as well as a breakdown by hospital site are shown in Table 1. Participants at SJH were older, and a higher proportion of SJH participants were in non-clinical roles. Breakdown of the clinical roles demonstrated an increased proportion of allied health professional participants at SJH. There were no differences between sites for sex or country of birth, while there were some differences in ethnic backgrounds across locations. Assessment of vaccination status demonstrated significant differences between sites, but post hoc testing identified no differences between the four vaccination statuses assessed. There were also no significant differences in vaccine types administered between locations, with the majority of participants receiving the Pfizer vaccine.

### 3.2. Variables Associated with Vaccination Status

Factors associated with vaccination status were assessed. Primary vaccination course with two subsequent booster vaccines had been received by n = 437 (35%), with n = 663 (53%) receiving primary vaccination course and one booster and n = 133 (11%) receiving only the primary vaccination course. There were n = 27 (2%) participants who had not received their primary vaccination course. There were significant differences in vaccination status across age and sex, with fully vaccinated participants (primary vaccination course and two boosters) more likely to be older and male when compared to those with incomplete boosted vaccine courses (r^2^ = 0.05, *p* < 0.001, r^2^ = 0.009, *p* < 0.01, respectively). Participants were also more likely to have received at least one booster dose if they were employed in a clinical role (r^2^ = 0.01, *p* < 0.01). The complete results are shown in Table 2. Differences within the vaccination status of those in clinical roles (n = 916) were investigated. Within clinical roles, n = 181 (96%) doctors/dentists, n = 358 (90%) nursing/midwifery, n = 43 (78%) healthcare assistants, n = 147 (90%) allied health professionals and n = 93 (84%) laboratory staff had completed their primary vaccination course and received two subsequent boosters. There were significant differences between these roles (r^2^ = 0.02, *p* < 0.001), with the proportion of doctors and dentists completing this course being greater than that of healthcare assistants (t = −3.59, *p* = 0.003) and laboratory staff (t = −3.35, *p* = 0.008). There were no differences between self-reported risks for SARS-CoV-2 acquisition between vaccination groups (r^2^ = 0.04, *p* = 0.08), while ethnicity and country of birth were also not associated with vaccination status.

An adjusted multivariate stepwise logistic regression model was used to evaluate whether the variables assessed on univariate analysis were independently associated with SARS-CoV-2 vaccination status. The significant univariate variables of age, sex, location and role (clinical versus non-clinical) were included in the model. Ethnicity, country of birth (Ireland vs. non-Ireland) and self-reported risk for COVID-19 were also included in the model. This demonstrated that increasing age, male sex, site of recruitment (SJH) and being employed in a clinical role were all independently associated with SARS-CoV-2 vaccine uptake among HCWs. The other variables included in the model did not meet the pre-specified threshold for statistical significance and were excluded from the final results. These results are shown in detail in Table 3. 

## 4. Discussion

Vaccination remains a cornerstone in the prevention of SARS-CoV-2 acquisition, reducing symptomatic disease, and limiting onward viral transmission. Engaging HCWs in ongoing vaccination programmes is challenging. Identification of HCW groups with poor engagement with vaccine programmes would allow for tailored and targeted campaigns, with the goal of increasing vaccine uptake across all HCW groups. This study uses demographic and professional factors to identify the HCW subgroups associated with lower vaccine uptake across two tertiary hospital sites on the island of Ireland. Importantly, we demonstrate that these subgroups exist independent of recruitment site, suggesting that these attitudes are inherent to the HCWs rather than being associated with a specific institutional factor. This allows for generalisation of our results beyond the study sites. We also identified site-specific differences in vaccine uptake. 

The significant differences between sites may be explained by the approach taken to vaccine administration. Both primary and booster SARS-CoV-2 vaccinations at SJH were offered on site, while vaccination at UHG were on site for the primary vaccination programme with booster doses administered at an off-site location. This change in vaccine access likely accounts for the differences in booster vaccination rates between sites, while there are no differences between sites for primary vaccination. Logistical issues and institutional obstacles to vaccine access have previously been shown to reduce vaccine uptake, both in the Irish context and internationally [26,27]. Future vaccination programmes should try and overcome these logistical obstacles by offering vaccination at the site of employment. 

We found that young female HCWs were less likely to be fully vaccinated against SARS-CoV-2. This result is in keeping with prior studies, where it was shown that older male HCWs had higher vaccination uptake [28]. These are not factors unique to SARS-CoV-2 vaccination, and have previously been reported in the context of influenza vaccination as well as immunity to measles, mumps and rubella [29,30]. Hesitancy among this population towards the SARS-CoV-2 vaccination has been previously linked with fears regarding fertility and reproductive health [31]. This is despite a wealth of studies demonstrating no adverse effects on pregnancy or fertility from SARS-CoV-2 vaccination [32,33]. Young adults also demonstrate increased infection rates when compared with the general population, and increases in young adult infections are associated with subsequent periods of peak incidence across the general population [34,35]. 

We demonstrate that HCWs in non-clinical roles are less likely to have up-to-date SARS-CoV-2 vaccinations, independent of age or sex. There are several possible reasons underlying low vaccine uptake in non-clinical workers. Non-clinical workers may have lower exposure to vaccine promotion campaigns, which have previously been associated with increased vaccine uptake [36]. The role of staff education across all staff grades regarding vaccination and the availability of accurate vaccine information should be considered in future vaccination programmes, especially as sharing of misbeliefs and misconceptions regarding vaccines are associated with reduced vaccine uptake [37]. 

It is notable that there were no ethnic disparities in vaccine uptake in our study. This is in contrast to lower uptake reported among minorities in other jurisdictions, predominantly the United States [38]. These disparities have been attributed to socioeconomic factors as well as prior experience of racial discrimination [39]. Our study suggests that these previously identified factors are not universally applicable across all healthcare systems. However, global HCW attitudes towards COVID-19 vaccination has been mixed. Vaccine hesitancy has been described across continents and across differing resource settings [40,41,42]. Indeed, a recent meta-analysis of HCW acceptance of COVID-19 vaccination examined studies from 13 different countries, and found that this was a highly controversial issue, with more than one third of study participants not in favour of HCW vaccine mandates [43]. The phenomenon of vaccine hesitancy is not unique to SARS-CoV-2. Vaccination against seasonal Influenza virus in HCWs is an ongoing challenge [44,45]. An important component of future pandemic preparedness will be a united global effort to combat vaccination distrust and improve HCW participation in vaccination programmes. 

The use of self-reported survey data is a potential limitation to this work. While vaccination status was independently confirmed via the national COVID-19 immunisation database, factors such as risks for COVID-19 infection were reliant on participant declaration. Given that participants were not mandated to enrol in the study, selection bias may have influenced the results, as well as non-response bias. Other forms of bias associated with similar study designs, such as acquiescence bias and answer option bias, were minimised by avoiding the use of leading questions and varying answer orders. Information and recall bias were mitigated by prospectively collecting data. The cross-sectional nature of this study also does not allow for direct causal inferences to be made, and there may be additional confounders accounting for the associations described. However, this study robustly investigated factors associated with HCW SARS-CoV-2 vaccine uptake, with a large study population across all HCW roles, capturing the duration of the SARS-CoV-2 vaccination programme in Ireland. There is a need to further study the reasons for reduced vaccine uptake in the identified cohorts. Qualitative descriptive research is required, exploring attitudes towards and understanding of SARS-CoV-2 infection and its associated risk across HCWs. Additionally, socio-economic information and educational attainment data would be valuable, as these have previously been associated with vaccine uptake [46,47]. Collection of these data was beyond the scope of the present study, as the rapid rollout and recruitment of a large number of participants precluded the use of extensive time-consuming questionnaires. 

The ongoing evolution of new circulating SARS-CoV-2 variants will likely require annual booster vaccination, in a similar manner to influenza. Our study demonstrates specific groups requiring targeted vaccination campaigns, in particular young females and workers in non-clinical roles. These data can be used to inform national vaccination programmes across healthcare systems, with the development of information campaigns targeted at these populations, increasing HCW vaccine uptake, improving HCW health, reducing patient risk and optimising absenteeism in the workplace.

## Figures and Tables

**Table 1 vaccines-11-01529-t001:** Cohort demographics.

	Total (n = 1260)	SJH (n = 640)	UHG (n = 620)	Statistic	*p* Value
Age; median (IQR)	43 (33–51)	44 (35–51)	41 (31–51)	t = 3.59	*p* = 0.0003
Sex, female; n (%)	998 (79)	515 (80)	483 (78)	Χ^2^ = 3.09	*p* = 0.21
Born in Ireland; n (%)	983 (78)	513 (80)	470 (76)	Χ^2^ = 3.47	*p* = 06
Ethnicity; n (%)				r^2^ = 0.02	*p* = 0.02
-White Irish	992 (79)	521 (81)	471 (76)	OR (95% CI)	
-Other White Irish	144 (11)	63 (10)	81 (13)	0.09 (0.001–0.2)	0.048
-Chinese	28 (2)	12 (2)	16 (3)	0.10 (−0.1–0.3)	0.31
-Other Asian	62 (5)	37 (6)	25 (4)	−0.07 (−0.2–0.1)	0.27
-Black African	14 (1)	3 (<1)	11 (2)	0.31 (0.05–0.6)	0.02
-Other	20 (2)	4 (<1)	16 (3)	0.33 (0.1–0.5)	0.004
Role; n (%)				r^2^ = 0.02	*p* < 0.001
-clinical	916 (73)	434 (68)	482 (78)	OR (95% CI)	
-non clinical	269 (21)	175 (27)	94 (15)	−0.18 (−0.24–−0.11)	<0.001
-unknown	75 (6)	31 (5)	44 (7)	0.06 (−0.06–0.18)	0.31
Clinical role; n (%)				r^2^ = 0.03	*p* < 0.001
-medical/dental	189 (15)	72 (11)	117 (19)	OR (95% CI)	
-nursing/midwifery	399 (32)	181 (28)	218 (35)	−0.07 (−0.2–0.01)	0.1
-HCA	55 (4)	26 (4)	29 (5)	−0.09 (−0.2–0.1)	0.23
-AHP	162 (13)	102 (16)	60 (10)	−0.25 (−0.4–−0.1)	<0.001
-Laboratory	111 (9)	53 (8)	58 (9)	−0.10 (−0.2–0.02)	0.1
Non-clinical role; n (%)				r^2^ = 0.03	*p* = 0.07
-administration	214 (17)	146 (23)	68 (11)	OR (95% CI)	
-porter/transport	18 (1)	10 (2)	8 (1)		
-technician	17 (1)	6 (<1)	11 (2)		
-security	5 (<1)	4 (<1)	1 (<1)		
-catering	16 (1)	10 (2)	6 (<1)		
Self-reported risk; n (%)	160 (13)	89 (14)	71 (11)	r^2^ = 0.003	*p* = 0.15
-biologic therapy	11 (<1)	7 (1)	4 (<1)		
-rituximab	1 (<1)	1 (<1)	0 (0)		
-methotrexate	5 (<1)	4 (<1)	1 (<1)		
-steroids	26 (2)	16 (3)	10 (2)		
-transplant	1 (<1)	0 (0)	1 (<1)		
-cancer	8 (<1)	2 (<1)	6 (<1)		
Vaccinations; n (%)				r^2^ = 0.009	*p* = 0.01
None	27 (2)	13 (2)	14 (2)	OR (95% CI)	
Primary	133 (11)	56 (9)	77 (12)	0.06 (−0.15–0.27)	*p* = 0.57
Primary & 1 booster	663 (53)	323 (50)	340 (55)	−0.01 (−0.20–0.19)	*p* = 0.95
Primary & 2 boosters	437 (35)	248 (39)	189 (30)	−0.09 (−0.30–0.11)	*p* = 0.38
Vaccine type; n (%)				r^2^ = 0.004	*p* = 0.25
Pfizer	987 (78)	504 (79)	483 (78)		
AstraZeneca	212 (17)	113 (18)	99 (16)		
Moderna	30 (2)	10 (2)	20 (3)		
Janssen/J+J	6 (<1)	3 (<1)	3 (<1)		
Other/unknown	25 (2)	10 (2)	15 (2)		

Definition of abbreviations: SJH = St James’s Hospital; UHG = University Hospital Galway; HCA = healthcare assistant; AHP = allied health professional; OR = odds ratio; CI = confidence interval, IQR = interquartile range; NS = not significant.

**Table 2 vaccines-11-01529-t002:** Variables associated with vaccination status.

	Primary & 2 Boosters (n = 437)	Primary & 1 Booster (n = 663)	Primary Only (n = 133)	Unvaccinated (n = 27)	Statistic
Age; median (IQR)	46 (39–54)	41 (31–49)	40 (32–48)	44 (25–49)	r^2^ = 0.05, *p* < 0.001 Primary/two boosters vs primary/one booster (t = 7.36, *p* < 0.001); primary/two boosters vs. primary only (t = 4.95, *p* < 0.001)
Sex, female; n (%)	323 (74)	541 (82)	112 (84)	22 (81)	r^2^ = 0.009, *p* < 0.01 Primary/two boosters vs primary/one booster (t = −2.89, *p* = 0.02); primary/two boosters vs. primary only (t = −2.82, *p* = 0.03)
Born in Ireland; n (%)	347 (79)	513 (77)	103 (77)	20 (74)	r^2^ = 0.001, *p* = 0.82
Located in SJH; n (%)	248 (57)	323 (49)	56 (42)	13 (48)	r^2^ = 0.01, *p* = 0.01 Primary/two boosters vs. primary only (t = −2.97, *p* = 0.02); primary/2 boosters vs primary/1 booster (t = −2.62, *p* = 0.045)
Ethnicity; n (%)					r^2^ = 0.001, *p* = 0.89
-White Irish	351 (80)	516 (78)	105 (78)	20 (74)	
-Other White Irish	46 (11)	76 (11)	18 (14)	4 (15)	
-Chinese	9 (2)	16 (2)	3 (<1)	0 (0)	
-Other Asian	19 (4)	38 (6)	2 (<1)	3 (11)	
-Black African	5 (<1)	6 (<1)	3 (<1)	0 (0)	
-Other	7 (2)	11 (2)	2 (<1)	0 (0)	
Role; n (%)					r^2^ = 0.01, *p* = 0.003 Primary/two boosters vs. primary only (t = −3.70, *p* = 0.001); primary/one booster vs. primary only (t = −3.28, *p* = 0.006)
-clinical	332 (76)	490 (74)	77 (58)	17 (63)	
-non clinical	83 (19)	132 (20)	45 (34)	9 (33)	
-unknown	22 (5)	41 (6)	11 (8)	1 (4)	
Clinical role; n (%)					r^2^ = 0.01, *p* = 0.03 No significant differences on post hoc analysis
-medical/dental	90 (21)	91 (14)	7 (5)	1 (4)	
-nursing/midwifery	123 (28)	235 (35)	35 (26)	6 (22)	
-HCA	11 (3)	32 (5)	10 (8)	2 (7)	
-AHP	64 (15)	83 (13)	11 (8)	4 (15)	
-Laboratory	44 (10)	49 (7)	14 (11)	4 (15)	
Non-clinical role; n (%)					r^2^ = 0.02, *p* = 0.15
-administration	73 (27)	102 (15)	32 (24)	7 (26)	
-porter/transport	2 (<1)	11 (2)	5 (4)	0 (0)	
-technician	5 (1)	7 (1)	3 (2)	2 (7)	
-security	1 (<1)	3 (<1)	1 (<1)	0 (0)	
-catering	2 (<1)	9 (<1)	4 (3)	1 (4)	
Self-reported risk; n (%)	69 (16)	72 (11)	18 (14)	1 (4)	r^2^ = 0.004, *p* = 0.08

Definition of abbreviations: SJH = St James’s Hospital; HCA = healthcare assistant; AHP = allied health professional; IQR = interquartile range, NS = not significant.

**Table 3 vaccines-11-01529-t003:** Multivariable model for factors associated with vaccination status.

	Association with Being Fully Vaccinated	
	Odds ratio (95% CI)	*p* value
Age	1.02 (1.01–1.03)	0.03
Sex (Female)	0.60 (0.38–0.94)	0.03
Location (UHG)	0.69 (0.49–0.98)	0.04
Role (Non-clinical)	0.59 (0.46–0.76)	<0.001

Stepwise logistic regression performed. Adjustments made for age, sex, location of employment, country of birth, ethnicity, self-reported risk for disease acquisition, and clinical vs. non-clinical role. Variables with *p* values > 0.20 were excluded from the results table. Definition of abbreviations: CI = confidence interval; UHG = University Hospital Galway.

## Data Availability

The data that support the findings of this study are available from the corresponding author upon reasonable request.

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
