# Peer review of "Healthcare Worker Characteristics Associated with SARS-CoV-2 Vaccine Uptake in Ireland; a Multicentre Cross-Sectional Study"

_vaccines, 2023, doi:10.3390/vaccines11101529_

Round 1

Reviewer 1 Report

This is an interesting study and will be helpful for the relevant field. However, the manuscript, in my opinion, does not have such content(s) so that it can be accepted in the journal Like 'Vaccines'. It is an survey and can be published in a related journal.     

Moderate checking

Author Response

We thank the reviewer for their comments. We feel that the manuscript in its updated form fits the scope of Vaccines, and this has been reflected in the comments received by other reviewers and by the Editorial Board for their consideration for publication in the first instance. 

Reviewer 2 Report

The manuscript (ID: vaccines-2620278) aimed to investigate engagement with SARS-CoV-2 vaccination amongst healthcare workers and identify characteristics associated with lower vaccine uptake in this population in Ireland.     

Comments (Major revision):       

  • Line 93: It is mandatory to introduce a new subsection `Study sample` and the following information:  

§  Specify the type of study sample.  

§  Which persons were eligible to participate in this study?  

§  What were the inclusion criteria for the respondents in this study?     

§  What were the criteria for exclusion from this study?    

§  What is the `Participation rate` and `Response rate` in this study?      

§  It is mandatory to specify whether the information on vaccination against COVID-19 was only based on the applied survey.  

§  It is mandatory to specify whether the information on vaccination against COVID-19 was verified by inspecting the medical documentation (such as a vaccination card).  

  • Lines 94-109: Define the `Self-reported risk` variable.

§  It is mandatory to specify whether the information on the `Self-reported risk` variable was only based on the applied survey.  

§  It is mandatory to specify whether the information on the `Self-reported risk` variable was verified by inspecting the medical documentation.    

  • Line 109: Add a new paragraph to provide an explanation for:

§  Dividing the `cohort` in this study by location (that is, SJH versus UGH).

§  Introducing the `Located in SJH` variable into univariate and multivariate analysis. 

  • Lines 110-122: Indicate for which variables the adjustment was made.   
  • Lines 124-126: Omussion? Correct this.    
  • Line 159: Add a description of statistical significance for the `Self-reported risk` variable.
  • Line 173: Explain why the `Self-reported risk` variable is not shown in Table 3, that is, why it was not included in the multivariate analysis model. Correct this. 
  • Lines 173-175: Under Table 3, list the variables for which the adjustment was made. 
  • Lines 220-232: Discuss various sources of bias (information bias, non-response bias, etc.) as potential limitations of this study.   

The quality of English language is appropriate. 

Reviewer 3 Report

I was invited to revise the paper entitled "Healthcare worker characteristics associated with SARS-CoV-2 vaccine uptake". It as a multicentre cross-sectional study of vaccine uptake and antibody response among HCWs from Ireland. The topic is interesting and can improve the knowledge on this topic. It is one of first paper from Ireland.

Observations:

- Title should be more informative: i suggest to add the information about setting. Exaample "Healthcare worker characteristics associated with SARS-CoV-2 vaccine uptake: a multicentre cross section study from Ireland";

- In introduction section Authors should report the vaccine schedule proposed in Ireland and the timeline of vaccine introduction in Ireland setting;

- In table 1 I suggest to report exact p-value for each comparisons;

- About methodology, vaccination adherence was a dichotomous variable so linar regression was not appropriate. Authors should perform logistic regression;

- In multivariable models Authors should add all variables and not only significant once. Authors can use a stepwise regressions;

- Among discussions, Authors should discuss the attitudes towards vaccination among HCWs from other countries.

Round 2

Reviewer 2 Report

Thank you for the opportunity to re-review manuscript ID: vaccines-2620278. The authors addressed all of my comments correctly, point by point. Accordingly, the authors made appropriate corrections and satisfactorily revised their manuscript. Also, the authors were very honestly about the issues of the limitations of this study. I believe that the corrections made by the authors in the revised version of this paper significantly contributed to making the paper more clear, transparent and informative. 

I thank the authors.    

The quality of English language is appropriate.